# A Framework for Four-Dimensional Variational Data Assimilation Based on Machine Learning

**DOI:** 10.3390/e24020264

**Published:** 2022-02-12

**Authors:** Renze Dong, Hongze Leng, Juan Zhao, Junqiang Song, Shutian Liang

**Affiliations:** 1College of Meteorology and Oceanography, National University of Defense Technology, Changsha 410000, China; dongrz20@nudt.edu.cn (R.D.); hzleng@nudt.edu.cn (H.L.); junqiang@nudt.edu.cn (J.S.); 2School of Resources and Environmental Engineering, Hefei University of Technology, Hefei 230000, China; 2021218341@mail.hfut.edu.cn

**Keywords:** numerical weather prediction, four-dimensional variational assimilation, machine learning, tangent linear and adjoint models

## Abstract

The initial field has a crucial influence on numerical weather prediction (NWP). Data assimilation (DA) is a reliable method to obtain the initial field of the forecast model. At the same time, data are the carriers of information. Observational data are a concrete representation of information. DA is also the process of sorting observation data, during which entropy gradually decreases. Four-dimensional variational assimilation (4D-Var) is the most popular approach. However, due to the complexity of the physical model, the tangent linear and adjoint models, and other processes, the realization of a 4D-Var system is complicated, and the computational efficiency is expensive. Machine learning (ML) is a method of gaining simulation results by training a large amount of data. It achieves remarkable success in various applications, and operational NWP and DA are no exception. In this work, we synthesize insights and techniques from previous studies to design a pure data-driven 4D-Var implementation framework named ML-4DVAR based on the bilinear neural network (BNN). The framework replaces the traditional physical model with the BNN model for prediction. Moreover, it directly makes use of the ML model obtained from the simulation data to implement the primary process of 4D-Var, including the realization of the short-term forecast process and the tangent linear and adjoint models. We test a strong-constraint 4D-Var system with the Lorenz-96 model, and we compared the traditional 4D-Var system with ML-4DVAR. The experimental results demonstrate that the ML-4DVAR framework can achieve better assimilation results and significantly improve computational efficiency.

## 1. Introduction

Numerical weather prediction (NWP) predicts future atmospheric states using numerical methods on high-performance computers to solve equations describing atmospheric dynamics and thermal processes under certain initial conditions. Hence, it can be seen as an initial value problem [1,2,3]. Information in the atmosphere is often expressed in the form of data. In order to obtain an accurate initial field, we need to increase the credibility of the data and artificially remove redundant information. This also means reducing entropy. Data assimilation (DA) merges observations with numerical model forecasts to estimate the current optimal atmospheric state. The analysis, which results from data assimilation, is employed as the initial field for NWP [4]. Four-dimensional variational assimilation (4D-Var) is the most popular data assimilation method, which is widely used in many operational NWP centers [5,6,7,8,9,10].

The calculation of the 4D-Var assimilation system depends on the forecast model and the functional minimization calculation process to a large extent [11]. The higher the accuracy of the forecast model, the better the effect of the assimilation system. However, the improvement of the model precision will not only increase the prediction time but also increase the computational cost of the tangent linear and adjoint models in the process of functional minimization [12]. Meanwhile, the real-time performance of operational forecast determines the importance of computational efficiency. Therefore, the realization of the 4D-Var system must take into account the improved accuracy of the forecast model in the case of ensuring calculation efficiency. Currently, the approach that is achieved by reducing the resolution of the model is often adopted in the operational 4D-Var assimilation systems.

The ensemble assimilation method is an alternative to 4D-Var [13]. Nevertheless, the ensemble method has a primary problem, which is that the number of ensemble members is much smaller than the dimensions of the system, resulting in sample errors, false correlations, and low-rank problems [14]. The difficulty of 4D-Var stems from the complex forecast model. The difficulty of 4D-Var can be cut down by reducing the complexity of the forecast model.

With the development of machine learning (ML), the application of ML has penetrated various fields. As a data-driven method, it does not care about the calculation of the traditional physical model but obtains the underlying features and law through training data and then gains the simulation results [15]. A great deal of forecast product data and satellite observations provide a good opportunity for the application of ML in earth science [16]. The ML method has achieved rich research results in the physical process simulation [17,18], parameter estimation, and DA [19].

Dueben and Bauer trained the deep neural network (DNN) using the reanalysis data on a coarse-resolution grid with a spatial resolution of 6 degrees. They employed the DNN to forecast 500 hPa geopotential height for global regions and demonstrated the feasibility of ML in the weather forecast [16]. Weyn et al. utilized the reanalysis data to train the convolutional neural network (CNN) and built a deep learning weather prediction (DLWP) to forecast the geopotential height of 500 hPa in the northern hemisphere and meteorological elements of 300–700 hPa [20]. The experiments show that the prediction accuracy of the DLWP for the geopotential height of 500 hPa is better than that of the T42IFS model and lower than that of the T63IFS model. In terms of computational efficiency, the running time of the DLWP is much lower than that of classical forecasting models, which proves that ML is an essential means to solve the problem of computational cost effectively. At present, the simulation accuracy of NWP for subgrid-scale physical processes needs to be improved. Furthermore, these small-scale processes will affect the accuracy of forecast results [21]. Therefore, it is crucial to improve the accuracy of the parameterization schemes of the physical process. Replacing traditional parameterization with ML is a way to improve accuracy. Rasp et al. used multilayer perceptron (MLP) to simulate a cloud parsing model. The experimental results show that the MLP parameterization scheme can run stably for a long time. Under the condition of ensuring the accuracy of the prediction results, the MLP parameterization scheme can reduce the computational cost [22]. Yuval et al. demonstrate that it is possible to add physical constraints to the neural network parameterization to improve the physical interpretability of the neural network parameterization scheme [23]. Song et al. used MLP to model the radiation parameterization scheme. The authors used the MLP parameterization scheme in the atmospheric model, which significantly reduced the root mean square error (RMSE) and increased the computational speed [24]. Krasnopolsky gave a detailed introduction to the prospects, methods, evaluation criteria, and limitations of neural networks in subgrid-scale physical processes [25]. Chantry et al. successfully emulated the nonorographic gravity wave drag scheme from the operational forecast model with the MLP [26]. The experimental results demonstrate that the emulator can be coupled to an operational system for seasonal timescales and is more accurate than the parameterized scheme used in operational predictions. Bonavita applied the artificial neural network (ANN) to simulate weak-constraint four-dimensional variational data assimilation (WC-4DVar) [27]. The results indicate that the assimilation products obtained by the ANN are similar to WC-4DVar. Furthermore, model errors can be corrected when the ANN is embedded in WC-4DVar. Hatfield et al. employed the MLP to simulate the parameterization of nonorographic gravity wave drag and applied the tangent linear and adjoint models of the MLP to 4D-Var [28]. The research demonstrates that the tangent linear and adjoint models of the MLP can be used for data assimilation and weather forecast. There is no significant difference between the assimilation forecast result of this method and the operational NWP center. Nonnenmacher takes advantage of the DNN to simulate the Lorenz-96 model and investigates whether the DNN derivatives are available [29]. The experimental results prove that the DNN can simulate kinetic models, and the accuracy of its derivatives is reliable and can be directly used for data assimilation and parametrization tuning.

Although ML has rich research results in the numerical forecast, most of these results are only for a single problem in the assimilation system, and it does not propose a pure data-driven data assimilation solution from a system-wide perspective. Based on the idea of ML simulator, this paper structures a 4D-Var assimilation system based on machine learing (ML-4DVAR). It replaces the two most time-consuming processes in the traditional 4D-Var system with machine learning: one is the forecast model, and the other is the tangent linear and adjoint models. In order to show the feasibility of the system, we conduct 4D-Var assimilation experiments with the Lorenz-96 model. The experiments demonstrate that the ML-4DVAR can get more accurate analysis results and improve computational efficiency compared to traditional implementations.

The remainder of the paper is organized as follows. Section 2 presents the structure of the ML-4DVAR. Section 3 investigates the performance of ML-4DVAR with the Lorenz-96 model. Finally, we conclude the results of this research and discuss future work in Section 4 and Section 5.

## 2. Methods

### 2.1. Related Knowledge

4D-Var utilizes the observations at different moments, the background at the initial moment, and the forecast model to obtain the analysis. The purpose of 4D-Var is to find an initial condition that makes the forecast trajectory to the greatest extent possible to fit the observation data in the interval [4]. As shown in Figure 1, the solid red line represents the predicted trajectory of the background, and the solid blue line represents the forecast trajectory of the analysis. The role of 4D-Var is to modify the forecast trajectory. We study a strongly constrained 4D-Var whose cost function is shown in Equation (Equation 1):(1)$$\begin{array}{cc} J\left(x_{0}\right) & =J_{b}+J_{o}\\ \; & =\frac{1}{2}{\left(x_{0}-x^{b}\right)}^{T}B^{-1}\left(x_{0}-x^{b}\right)\\ \; & +\frac{1}{2}\sum_{i=0}^{N}{\left[H_{i}\left(M_{i}\left(x_{0}\right)\right)-y_{i}^{o}\right]}^{T}R_{i}^{-1}\left[H_{i}\left(M_{i}\left(x_{0}\right)\right)-y_{i}^{o}\right] \end{array}$$
where $x_{0}$ is the control variable, $x^{b}$ denotes the background, $y_{i}^{o}$ represents the observations at time *i*, B is the background error covariance matrix, the significance of $R_{i}$ is the observation error covariance matrix at time *i*, $H_{i}$ represents the observation operator at time *i*, and $M_{i}$ is the forecast model at time *i*. It can be seen from Equation (Equation 1) that the cost function *J* is composed of two items, The first term represents the square of the deviation between the control variable and the background; the second term is the sum of the squares of the differences between the model integrated and the observations.

### 2.2. Problem Statement

The output of the 4D-Var is called analysis, which is denoted by $x^{a}$. Generally, the analysis is gained employing the quasi-Newton iteration method or the conjugate gradient method to calculate the minimum value of the cost function *J*. During the calculation, we need to integrate the forward forecast model and then figure the gradient of the cost function with respect to the control variable. At present, there are mainly two methods to compute the gradient: one is the finite difference method, and the other is the adjoint method. The finite difference method cannot guarantee the computer precision, and the amount of calculation is too expensive. The adjoint method for calculating gradients has two advantages: one is the small amount of computation, and the other is the high computational accuracy. Therefore, the operational systems generally take advantage of the adjoint method to calculate the gradients, which requires the tangent linear and adjoint models. The tangent linear and adjoint models are obtained by linearizing the nonlinear model. Most atmospheric models are highly nonlinear, including some unresolved parameter schemes. It often takes much time to integrate these models, and the tangent linear and adjoint models of these modes are complicated [13]. In 4D-Var, the forecast models are also essential. Usually, we need to spend a lot of time integrating forecast models, and tangent linear and adjoint models are closely related to the forecast models. These problems have seriously affected the performance of 4D-Var, which in turn affected the quality and timeliness of the weather forecast. With the rapid development of ML, the conditions for the application of ML to earth science are gradually maturing. We employ ML research to address the above problems.

### 2.3. The Architecture of ML-4DVAR

The key to constructing ML-4DVAR is to build the ML model to simulate the numerical prediction model. This precondition requires us to study the equations of the dynamical system. Ordinary differential equations (ODEs) are often applied to denote, understand, and predict the systems that change over time. Their basic form is shown in Equation (Equation 2):(2)$$\frac{dx(t)}{dt}=f\left(t,x\left(t\right)\right).$$

For the purpose of predicting the future, we need to integrate ODEs. Given the time step dt, the state value at time *i* + 1 is:(3)$$x_{i+1}=F\left(x_{i}\right)=x_{i}+\int_{i}^{i+1}f\left(x_{i}\right)dt.$$

It can be seen from Equation (Equation 3) that the relationship between $x_{i+1}$ and $x_{i}$ can be regarded as a functional relationship (or as a mapping from $x_{i}$ to $x_{i+1}$), where the independent variable is $x_{i}$ and the dependent variable is $x_{i+1}$. The operatinal systems are hard to compute mathematically analytical solutions to ODEs, so these equations need to be solved by numerical methods after discretization in time and space [1]. The formula is shown in Equation (Equation 4):(4)$$x_{i+1}=M\left(x_{i}\right)+\epsilon_{i}$$
where M describes the forecast model, and $\epsilon_{i}$ is the forecast model error.

Neural networks are a branch of ML. Neural networks can precisely simulate complex systems [25], and Vapnik has demonstrated that shallow neural networks can fit any function [30]. In theory, the neural network can fit any function [31], so we apply the neural network to simulate function $x_{i+1}=F\left(x_{i}\right)$. The conventional neural networks include a convolutional layer, pooling layer, fully connected layer, batch normalization layer, and nonlinear activation function [32]. There are nonlinear calculation processes in the dynamic system. These nonlinear computational processes may lead to the poor simulation of traditional neural networks [33]. Compared to traditional neural networks, the bilinear neural networks (BNNs) with bilinear layers can better simulate dynamical systems and are physically easier to explain. In this paper, a BNN is used to simulate the operator *f* to obtain $\hat{f}$ (where $\hat{f}$ represents the neural network operator), and then, the forecast model based on the BNN is established. The neural network forecast model used in this paper is shown in Figure 2, where the input value is $x_{i}$ and the output value is $x_{i+1}$. The BNN includes two convolutional layers and one bilinear layer, and the convolution kernels of the two convolutional layers are 4 and 1, respectively. We give dt, enter $x_{i}$, and then integrate simulation operator $\hat{f}$ using the fourth-order Runge–Kutta method to gain $x_{i+1}$.

The definition of the symbols in Figure 2 is as follows:(5)$$\begin{array}{cc} S_{1} & =\hat{f}\left(x_{i}\right)\\ S_{2} & =\hat{f}\left(x_{i}+\frac{dt}{2}S_{1}\right)\\ S_{3} & =\hat{f}\left(x_{i}+\frac{dt}{2}S_{2}\right)\\ S_{4} & =\hat{f}\left(x_{i}+dtS_{3}\right)\\ \hat{f}\left(x\right) & =\frac{dx}{dt}\\ x_{i+1} & =x_{i}+\frac{dt}{6}\left(S_{1}+2S_{2}+2S_{3}+S_{4}\right). \end{array}$$

The training data are generated by integrating the ODEs, and the integration method used is the fourth-order Runge–Kutta integration method. $\hat{M}$ represents the neural network forecast model, $x_{i}$ is the input variable of $\hat{M}$, $x_{i+1}$ is the output variable of $\hat{M}$, and its output is $x_{k+1}$. The cost function is defined as shown in Equation (Equation 6), and the optimization algorithm is Adam.
(6)$$L\left(W\right)=∥x_{i+1}-\hat{M}\left(x_{i}\right){∥}_{2}$$
where ${∥\cdot ∥}_{2}$ represents 2-norm.

After acquiring the BNN, this article uses the BNN and the tangent linear and adjoint models of the BNN to build ML-4DVAR. The flow of ML-4DVAR is shown in Figure 3. The following is an explanation of the flowchart. $x^{b}$, $y_{i}^{o}$, and $x^{a}$ have the same meaning as before, and $x_{i}$ stands for the background forecast at the ith observations time in the assimilation time window. There are a total of N + 1 observations in the assimilation time window (i,i+1). The process of ML-4DVAR is mainly divided into the following steps:①At the start time i of the assimilation time window, the previous forecast is regarded as the initial field. After the initial field is gained, the NN model forecasts until the end time i + 1 of the assimilation time window. The forecast obtained in this step is called the background forecast.②The cost function is computed. The cost function is the sum of the model observation equivalents and the observations difference in the assimilation time window. The model observation equivalents are the output of the observation operator acting on the background forecast.③The gradient of the cost function with respect to the control variable is calculated, and the calculation of the gradient requires the help of the tangent linear and adjoint models of NN.④We use an appropriate optimization algorithm to estimate the correction value of the state variable.⑤Return to ①; the following optimization cycle is started and runs until it meets the accuracy requirements and stops, and $x^{a}$ is output.⑥The forecast field at time i + 1 is calculated, the initial field is $x^{a}$, and the forecast model is NN, and then, the next analysis cycle begins.

As shown in Figure 3, the forecast model and tangent linear and adjoint models of ML-4DVAR are all derived from the neural network. The operation of the assimilation system does not rely on the physical model but entirely on the neural network.

## 3. Experiments and Results

In this section, we mainly introduce the Lorenz-96 model, the simulation effect of the BNN on the Lorenz-96 model, and the comparison results between various assimilation systems. We introduce Original-4DVAR, Joint-4DVAR, and ML-4DVAR that appear in the experiment. Original-4DVAR is a traditional 4D-Var assimilation system, and its forecast model and tangent linear and adjoint models are entirely derived from the physical model, that is, the Lorenz-96 model. Joint-4DVAR is the joint 4D-Var assimilation system, its forecast model is from Lorenz-96, and the tangent linear and adjoint models are from the BNN model. ML-4DVAR is a 4D-Var assimilation system based on ML, and its forecast model and tangent linear and adjoint models are derived from the BNN model.

### 3.1. Lorenz-96 Model

Atmospheric systems are extremely nonlinear, which means they have a high degree of complexity, and the amount of code for their numerical models is enormous. In the research process, the new methods are directly tested on the NWP, their computational cost is usually high, and it is not easy to obtain the test results in a short time [34]. For these reasons, researchers often use simplified models to test new methods. For example, Lorenz studied predictability on low-order systems, and his research results broke the notion that deterministic systems are entirely predictable; Platzman researched truncated spectral models on the Burgers equations, laying the foundation for the application of spectral models in operational systems [35]. The model used in this article is a nonlinear chaotic dynamic system named the Lorenz-96 model [36,37]. In the research of data assimilation, the Lorenz-96 model is often used as a test model by researchers [38,39]. The definition of the Lorenz-96 model is as described in Equation (Equation 7). The model contains the main characteristics of atmospheric motion: the first term on the right side represents the advection term, the second term is the dissipation term, and the meaning of the third item indicates external coercion.
(7)$$\frac{dx_{j}}{dt}=\left(x_{j+1}-x_{j-2}\right)x_{j-1}-x_{j}+F,j=1,2,\ldots ,J$$
where *j* represents the grid point coordinates, *F* is the external forcing parameter, and the significance of $x_{j}$ is the state variable of the model. The Lorenz-96 model adopts periodic boundary conditions, which are specifically expressed as $x_{-1}=x_{J-1},x_{0}=x_{J},x_{J+1}=x_{1}$, J≥4. In this article, we set J=40 and F=8. The reason for setting F=8 in this paper is that the system is in a state of chaos under this external forcing.

### 3.2. Performance of the Neural Network Forecast Model

In minimizing the cost function of 4D-Var, it is necessary to calculate the gradient of the cost function with respect to the control variable. The prerequisite for this purpose is that the neural network model can precisely simulate the Lorenz-96 model. This article compares the BNN and the CNN used by Seiya. Seiya employed a traditional neural network CNN to simulate the Lorenz-96 model [40]. This experiment aims to select a neural network with excellent simulation results. This article uses MSE as the cost function, so the RMSE of the predicted value and the actual value is adopted as the evaluation index. The initial values input to the BNN and CNN are the same. The two models predict 100 steps forward, and the time step dt=0.05 model time unit (MTU). The results are shown in Figure 4, where the solid blue line represents the RMSE of the CNN, and the solid yellow line represents the RMSE of the BNN. It can be seen from the figure that over time, the BNN has a more prominent advantage in reducing RMSE than the CNN. When the two models run to the 20th time step, the RMSE of the BNN is 0.010501, the RMSE of the CNN is 0.237361, the RMSE of the BNN is 95.6% lower than that of CNN under the same conditions. It can be seen from the experimental results that the error between the predicted value and the real value will increase rapidly at the beginning. When it reaches a particular moment, the increase of the error will slow down until the error stabilizes. For a period of time, the neural network can simulate dynamical systems. After analyzing the experimental results, we found that the simulation effect of BNN was better, so we chose BNN to simulate the Lorenz-96 model.

In order to further observe and analyze the simulation performance of the BNN, we plotted the distribution of the predicted values and the true values over 100-time steps. As shown in Figure 5, Figure 5a is the distribution of the BNN predicted values in time and space, Figure 5b is the distribution of real values in time and space, and Figure 5c is the distribution of difference values. Before the 20th time step, both errors are minimal in each component. After the 20th time step, the error is gradually obvious. The error increases and then oscillates around the maximum value in this process.

### 3.3. The Cost Function Settings

4D-Var needs to construct a cost function. In this article, the cost function used by Original-4DVAR is in the form of Equation (Equation 2). The background error covariance matrix B is calculated using the NMC method, and the calculation formula of the NMC method is shown in Equation (Equation 8). In the NMC method, the structure of B is the average of the difference between many (for example, 50) two different short-term forecasts at the same time, and the magnitude of B is appropriately scaled. In this article, λ is the scale parameter. The observation error covariance matrix $R_{i}=0.5I$, and the observation operator $H_{i}=I$. The length of the assimilation time window is 0.05 MTU. There are four observations in each assimilation time window, and the time interval of each observation is equal, which is 0.0125 MTU.
(8)$$B\approx \lambda E\left\{\left[x^{f}\left(48h\right)-x^{f}\left(24h\right)\right]{\left[x^{f}\left(48h\right)-x^{f}\left(24h\right)\right]}^{T}\right\}$$

The cost function form of 4D-Var employing the tangent linear and adjoint models of the BNN is similar to Equation (Equation 2), but the background error covariance matrix B is different. The B is set to αI. The observation error covariance matrix and observation operator are the same as in Original-4DVAR. This experiment utilizes different α to test the assimilation effect, and the results obtained are shown in Figure 6. When α=1, the RMSE is the largest, and its value equals 0.387314; when α=0.01, the RMSE is the smallest, and its value is equal to 0.170927, the difference between the two is 0.216387. When α is in the range of [0, 0.1], the RMSE changes very little. In this interval, the maximum RMSE only increases by 0.22% compared to the minimum RMSE. When α=0.01, RMSE achieves the minimum value, so the next experiment in this article chooses the cost function when α=0.01, and its form is shown in Equation (Equation 9).
(9)$$\begin{array}{cc} J\left(x_{0}\right) & =J_{b}+J_{o}\\ \; & =\alpha \frac{1}{2}{\left(x_{0}-x^{b}\right)}^{T}I^{-1}\left(x_{0}-x^{b}\right)\\ \; & +\frac{1}{2}\sum_{i=0}^{N}{\left[H_{i}\left(M_{i}\left(x_{0}\right)\right)-y_{i}^{o}\right]}^{T}R_{i}^{-1}\left[H_{i}\left(M_{i}\left(x_{0}\right)\right)-y_{i}^{o}\right] \end{array}$$

### 3.4. Evaluation

The evaluation indicators selected in this paper are root mean square error (RMSE), determinable coefficient ($R^{2}$), and Nash–Sutcliffe model efficiency (NSE), which are used to evaluate the assimilation forecast performance of the system. The selection of these indicators is based on the evaluation indicators used by Lei et al. when evaluating the air temperature data products of the Global Land Data Assimilation System (GLDAS) [41].
The root mean square error (RMSE) is the square root of the ratio of the square of the difference between the two datasets to the number of observations [42]. RMSE signifies the total error between the two datasets. The overall errors are the constitution of two errors: the first part of errors are systematic errors, and the second part of errors are unsystematic errors. The value range of RMSE is [0,+∞). The closer the RMSE is to 0, the smaller the difference between the two datasets. The definition of RMSE is shown in Equation (Equation 10).
(10)$$\mathrm{RMSE}={\left[\frac{1}{n}\sum_{i-1}^{n}{\left(x_{i}-y_{i}\right)}^{2}\right]}^{1/2}$$Determinable coefficient ($R^{2}$) is a statistic that measures the goodness of fit [43]. $R^{2}$ is the ratio of the covariance of the two datasets to the standard deviation of the two datasets. The value range of $R^{2}$ is [0,1]. The closer $R^{2}$ is to 1, the stronger the correlation between the two datasets. The definition of $R^{2}$ is shown in Equation (Equation 11).
(11)$$R^{2}={\left[\frac{\sum_{i=1}^{n}\left(x_{i}-\bar{x}\right)\left(y_{i}-\bar{y}\right)}{\sqrt{\sum_{i=1}^{n}{\left(x_{i}-\bar{x}\right)}^{2}}\sqrt{\sum_{i=1}^{n}{\left(y_{i}-\bar{y}\right)}^{2}}}\right]}^{2}$$Nash–Sutcliffe model efficiency (NSE) is often employed to quantify the prediction accuracy of simulation models (such as hydrological models). It can be used to express the accuracy of model output results [44]. NSE is obtained by subtracting the mean squared error of the target dataset and the standard dataset to the variance of the standard dataset from one. The value range of NSE is (−∞,1]. The closer the NSE value is to 1, the better the predictive ability of the model and the higher the consistency between the target dataset and the standard dataset. Its definition is shown in Equation (Equation 12).
(12)$$\mathrm{NSE}=1-\frac{\sum_{i=1}^{n}{\left(x_{i}-y_{i}\right)}^{2}}{\sum_{i=1}^{n}{\left(x_{i}-\bar{x}\right)}^{2}}$$
where $x_{i}$ and $y_{i}$ represent the value on the *i* grid point, $\bar{x}$ and $\bar{y}$ signify the average value of x and y, the dataset x represents standard data. In this article, $x^{t}$ is used as the standard data, and $x^{a}$ or $x^{f}$ is used as the target data.

### 3.5. 4D-Var Experiments

We tested the performance of the newly built 4D-Var and compared the Original-4DVAR, Joint-4DVAR, and ML-4DVAR. The observations required by these three systems are the same, and they are all acquired by adding disturbances to the real values; the disturbances follow a Gaussian distribution with mean 0 and variance 0.5, as shown in Equation (Equation 13). The real values are the solutions of the Lorenz-96 model at each moment under given initial conditions.
(13)$$\begin{array}{cc} y_{i}^{o} & =x_{i}^{t}+\sigma \\ \sigma & ∼N(0,0.5) \end{array}$$
where σ represents the disturbances.

#### 3.5.1. The Joint-4DVAR

In order to better compare the assimilation performance of Joint-4DVAR, we computed the RMSE, $R^{2}$ and NSE of $x^{a}$ of Original-4DVAR and Joint-4DVAR. The assimilation results of Joint-4DVAR and Original-4DVAR are shown in Figure 7. The solid yellow line represents the RMSE of $x^{a}$ of Original-4DVAR, and the solid blue line represents the RMSE of $x^{a}$ of Joint-4DVAR. Figure 7a shows the RMSE at each analysis time, Figure 7b shows the $R^{2}$ at each analysis time, and Figure 7c shows the NSE at each analysis time. It can be seen from the figure that at each analysis moment, the RMSE of Joint-4DVAR is less than the RMSE of Original-4DVAR, and the $R^{2}$ and NSE of Joint-4DVAR are greater than the $R^{2}$ and NSE of Original-4DVAR. The results in the figure qualitatively show that the assimilation effect of Joint-4DVAR is better than that of Original-4DVAR. As shown in the figure, RMSE, $R^{2}$, and NSE rose rapidly during the first period and stabilized after the 30th time step. This phenomenon is because the assimilation system, the background, and the observation need to be run in when the assimilation system is just started. This period is also called the start-up time. During the spin period, the results of the assimilation system are not available.

In order to quantitatively compare the assimilation effects of Joint-4DVAR and Original-4DVAR, the average values of RMSE, $R^{2}$, and NSE are recorded in Table 1. The three indicators are the average from the 50th analysis time to the 1000th analysis time. As can be seen from the table, Joint-4DVAR compared with Original-4DVAR, RMSE is reduced by approximately 43.9%, $R^{2}$ is approximately increased by 0.5%, and NSE increases by approximately 0.5%. The RMSE of Joint-4DVAR is the smallest. The results demonstrate that the overall error of Joint-4DVAR is the smallest, and the difference between $x^{a}$ and $x^{t}$ of Joint-4DVAR is 0.171967. Joint-4DVAR has the largest $R^{2}$. The results show that $x^{a}$ and $x^{t}$ of Joint-4DVAR have the highest degree of fit. If $x^{a}$ and $x^{t}$ are put into the regression model, 98.0741% of the fluctuation of $x^{t}$ can be explained by $x^{a}$. Joint-4DVAR has the largest NSE. The results indicate that the $x^{a}$ and $x^{t}$ of Joint-4DVAR have the best consistency.

The short-term forecast $x^{f}$ is usually also used to test the assimilation effect. $x^{f}$ is the result of forecasting employing $x^{a}$. The accuracy of $x^{f}$ is not only related to the assimilation module of the system but also depends on the performance of the forecasting module. The comparison results of $x^{f}$ of Joint-4DVAR and Original-4DVAR are displayed in Table 2. The results in Table 2 are the average values from the 50th time step to the 1000th time step. It can be seen that compared with Original-4DVAR, Joint-4DVAR has the most significant change in RMSE, which is approximately a decrease of 39.7%, while $R^{2}$ is approximately 0.4%, and NSE is approximately 0.5%.

The computational efficiency of the numerical prediction system is crucial. In NWP, data assimilation accounts for about 50% of the total running time [28]. While improving the accuracy of assimilation results, we must also pay attention to the time spent in assimilation. The time taken for Joint-4DVAR and Original-4DVAR to run for 1000 time steps is shown in Table 3. It can be seen from the table that the running time of Joint-4DVAR is significantly shorter than that of Original-4DVAR. The running time of Joint-4DVAR is reduced by 479.174939 s compared with the running time of Original-4DVAR, and the reduced running time accounts for about 65.9% of the running time Original-4DVAR.

The above-mentioned experimental results are the average of 50 experiments, and each experiment has been carried out for 1000 analysis cycles to avoid chance. It can be seen from the results that the assimilation performance of Joint-4DVAR is better than that of Original-4DVAR. Joint-4DVAR not only makes the assimilation result from $x^{a}$ closer to the true $x^{t}$; it also makes the forecast result in $x^{f}$ more accurate. Joint-4DVAR can run stably for a long time, and the system’s stability is trustworthy. The calculation efficiency of Joint-4DVAR is higher than that of Original-4DVAR. This shows that the running time of the assimilation module of Joint-4DVAR is less than that of Original-4DVAR. Through experiments and analysis of experimental results, we can find that the assimilation performance of Joint-4DVAR is superior to that of Original-4DVAR, and the calculation efficiency of Joint-4DVAR is more efficient than that of Original-4DVAR.

#### 3.5.2. The ML-4DVAR

ML-4DVAR and Original-4DVAR are different in two ways. The first is the neural network forecast module, and the second is the 4D-Var assimilation module built using the tangent linear and adjoint models of the BNN. We calculated and plotted the RMSE, $R^{2}$, and NSE of $x^{a}$ of ML-4DVAR at each moment. As shown in Figure 8, the solid yellow line represents Original-4DVAR, and the solid blue line represents ML-4DVAR. Figure 8a shows the RMSE at each time, Figure 8b shows the $R^{2}$ at each time, and Figure 8c shows the NSE at each time. It can be seen from the figure that at each moment, the RMSE of ML-4DVAR is smaller than that of Original-4DVAR, and the $R^{2}$ and NSE of ML-4DVAR are larger than those of Original-4DVAR.

The RMSE, $R^{2}$, and NSE of ML-4DVAR and Original-4DVAR are recorded in Table 4. It can be seen from the table that compared with Original-4DVAR, ML-4DVAR has an approximately 44.5% reduction in RMSE, approximately 0.5% increase in $R^{2}$, and approximately 0.5% increase in NSE. It can be seen from the table that compared with Original-4DVAR, the RMSE of $x^{f}$ of ML-4DVAR is reduced by 42.5%, $R^{2}$ is increased by about 0.4%, and NSE is increased by about 0.5%. The running time of ML-4DVAR is 569.110456 s lower than that of Original-4DVAR, which accounts for about 78.3% of the running time of ML-4DVAR. These experimental results indicate that $x^{a}$ and $x^{f}$ of ML-4DVAR are closer to $x^{t}$ than $x^{a}$ and $x^{f}$ of Original-4DVAR. The running time of ML-4DVAR is shorter than that of Original-4DVAR.

This paper compares the assimilation performance and computational efficiency of ML-4DVAR and Joint-4DVAR. The assimilation and computing performance of ML-4DVAR and Joint-4DVAR are shown in Table 5. It can be seen that the assimilation performance of ML-4DVAR is better than that of Joint-4DVAR, and the running time of ML-4DVAR is less than that of Joint-4DVAR. The assimilation performance of ML-4DVAR is only slightly better than that of Joint-4DVAR. We compare the three indicators of $x^{a}$. Compared with Joint-4DVAR, the RMSE of ML-4DVAR is reduced by 1.2%, and the increase in $R^{2}$ and NSE is less than $10^{-4}$. The running time of ML-4DVAR is 36.2% less than that of Joint-4DVAR. These results show that the neural network prediction model can significantly improve the computational efficiency of the numerical prediction system.

#### 3.5.3. The ML_O_-4DVAR

In the real world, the real model is not available, making it impossible to train neural networks using the data generated by the real model. Although the real model data are not available, the observation data are available. The observations can generally be regarded as the real values added with disturbance. In data assimilation, it is assumed that the disturbance follows a normal distribution. In order to be close to the real situation, this article uses observation data as training data, trains the neural network model, and then tests the performance of the neural network model. We built an assimilation system on the trained neural network model. The system is named ${\mathrm{ML}}_{O}$-4DVAR, where the subscript O represents the neural network model trained on observation data. The RMSE, $R^{2}$, and NSE of $x^{a}$ at each moment are shown in Figure 9.

In the first period, the assimilation effect of ${\mathrm{ML}}_{O}$-4DVAR is not as good as that of Original-4DVAR. After the 50th time step, the assimilation effect of ${\mathrm{ML}}_{O}$-4DVAR is better than that of Original-4DVAR. Since the neural network model used by ${\mathrm{ML}}_{O}$-4DVAR is obtained by training observation data, it has not reached stability during the spin-up period. So, in the first period, the effect of ${\mathrm{ML}}_{O}$-4DVAR was not very good. The results of the assimilation performance and computational efficiency of ${\mathrm{ML}}_{O}$-4DVAR and Original-4DVAR are recorded in Table 6. The results in the table are the average value from the 50th time step to the 1000th time step. We compare RMSE, $R^{2}$, and NSE of ${\mathrm{ML}}_{O}$-4DVAR and Original-4DVAR $x^{a}$. Compared with Original-4DVAR, the RMSE of ${\mathrm{ML}}_{O}$-4DVAR has been reduced by 30.4%, and $R^{2}$ and NSE have increased by approximately 0.4%. The calculation efficiency of ${\mathrm{ML}}_{O}$-4DVAR is 76.1% higher than that of PHY-NPS. The assimilation effect and calculation efficiency of ${\mathrm{ML}}_{O}$-4DVAR are better than those of Original-4DVAR. ${\mathrm{ML}}_{O}$-4DVAR and ML-4DVAR are numerical prediction systems based on neural networks. The difference between them is the difference in training data. It can be seen from Table 7 that ML-4DVAR has the best assimilation performance and computational efficiency. The above experimental results demonstrate that although the performance of the neural network model trained with observation data is not the best, it is available.

## 4. Discussion

In the study, we make use of the tangent linear adjoint models of the ML model in 4D-Var. The prerequisite for applying the tangent linear and adjoint models of the ML model in 4D-Var is that the ML model accurately simulates the physical model. First, the BNN was built according to the characteristics of the physical model. Then, Joint-4DVAR is established, in which the tangent linear and adjoint models are derived from the BNN, and the prediction model is derived from the physical model. After that, this article tests the performance and computational efficiency of ML-4DVAR. ML-4DVAR is an assimilation system based on the ML model. Its prediction model and tangent linear and adjoint models are all provided by the ML model. Finally, we train the ML model on the observation data and build the 4D-Var assimilation system on this basis. The above results are discussed as follows:

The BNN trained on Lorenz-96 model data can simulate the Lorenz-96 model. The RMSE of the one-step predicted value and the actual value of the BNN is 4.46 × $10^{-4}$, it can be seen that the RMSE is very small. The consequences indicate that BNN can simulate and predict dynamic systems well. The reason is that the bilinear operation embedded in the neural network is an essential feature of the dynamic system [33].

The Joint-4DVAR is reliable, and its computational efficiency is satisfactory. Through the analysis of the experimental results, we can see that the overall error between the Joint-4DVAR analysis and the forecast and the true is more minor. The forecast models of Joint-4DVAR and Original-4DVAR are derived from physical models. The assimilation module of Joint-4DVAR is different from that of Original-4DVAR. The 4D-Var used in the assimilation module of Joint-4DVAR is built based on the tangent linear and adjoint models of the neural network. The 4D-Var employed by the assimilation module Original-4DVAR is established based on the tangent linear and adjoint models of the physical model. The performance of Joint-4DVAR is better than that of Original-4DVAR, and the calculation efficiency is higher, indicating that the performance and calculation efficiency of the assimilation module of Joint-4DVAR is higher than that of Original-4DVAR. The results show that the tangent linear and adjoint models of the neural network can be used in 4D-Var, and its calculation results are more accurate, and the running time is shorter.

This article builds ML-4DVAR on the Lorenz-96 model. It can be seen from the experimental results that the performance of ML-4DVAR is better than that of Original-4DVAR, and the computational efficiency of ML-4DVAR is higher. This article also compares the assimilation performance and computational efficiency of ML-4DVAR and Joint-4DVAR. We can see that compared with Joint-4DVAR, the assimilation performance of ML-4DVAR is improved very little, while the computational efficiency of ML-4DVAR is greatly improved. The assimilation modules of ML-4DVAR and Joint-4DVAR are the same, and their prediction modules are different. ML-4DVAR uses the neural network models for prediction, and Joint-4DVAR utilizes the physical models. The calculation efficiency of ML-4DVAR is higher than that of Joint-4DVAR. The result shows that neural networks can accelerate the forecasting process.

Neural networks trained using observation data are available. Although ${\mathrm{ML}}_{O}$-4DVAR was not very stable during the first period, after the 50th time step, ${\mathrm{ML}}_{O}$-4DVAR can be employed for assimilation and forecasting. We compared Joint-4DVAR, ML-4DVAR, and ${\mathrm{ML}}_{O}$-4DVAR. The assimilation performance and computational efficiency of ML-4DVAR are the best. This results indicate that the pure data-driven numerical prediction system is feasible in the Lorenz-96 model.

In summary, the BNN can simulate dynamic models well. The performance of Joint-4DVAR is excellent, which shows that the physical model and the 4D-Var based on the tangent linear and adjoint models of the ML model can work together. Among the three assimilation systems, Original-4DVAR, Joint-4DVAR, and ML-4DVAR, the system with the best assimilation performance and calculation efficiency is ML-4DVAR. The results prove that the assimilation system composed of the ML model and its tangent linear and adjoint models are satisfactory. This paper establishes the 4D-Var assimilation system based on ML. This study provides a method to obtain the tangent linear and adjoint models in 4D-Var.

## 5. Conclusions

In order to reduce the development difficulty of the tangent linear adjoint model and improve the computational efficiency of 4D-Var, we establish ML-4DVAR. ML-4DVAR’s forecast model and tangent linear and adjoint models are derived from the ML model. The experiments show that the assimilation performance and computational efficiency of ML-4DVAR are better than those of Original-4DVAR. The results prove that building the 4D-Var assimilation system based on ML is feasible. This study shows that the forecast model based on the ML model and the Jacobians of the ML model can work stably for a long time in 4D-Var. This study expands the application scope of neural networks in NWP and provides a reference for the future combination of ML and DA.

However, there is still a problem in this study. From the experimental results, it can be seen that the results of ML-4DVAR are not available in the first 50 steps of the system just running. In the future, we need to improve and perfect the assimilation performance of the assimilation system in the early stage.

Nowadays, with the generation of large amounts of data and the emergence of various open-source software, we can build ML models more simply. Building a ML model is cheaper and faster than a physical model. There are two main applications of ML in NWP: one is to improve the accuracy of weather forecast [26], and the other is to improve calculation efficiency. We need to build appropriate the ML models for different problems in this process. This method can reduce the difficulty of developing tangent linear and adjoint models, thereby expanding the application range of 4D-Var. The ultimate goal is to improve the accuracy of weather forecast in order to better understand and predict atmospheric systems. In the future, we need to build a suitable ML model for the actual atmospheric model in the future to support the application of ML in numerical weather prediction.

## Figures and Tables

**Figure 1 entropy-24-00264-f001:**
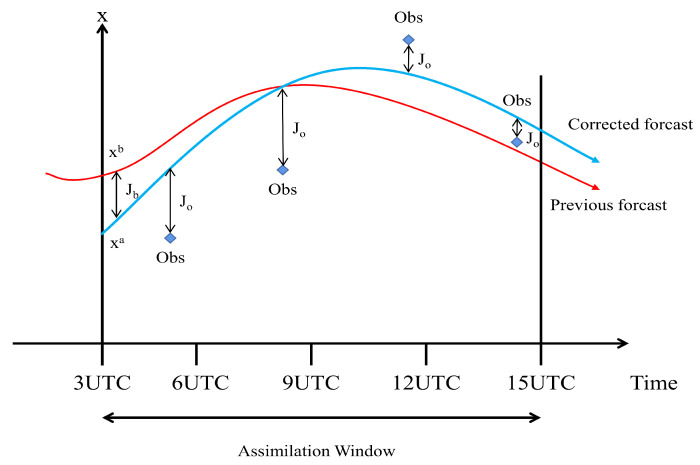
4D-Var assimilation in the NWP.

**Figure 2 entropy-24-00264-f002:**
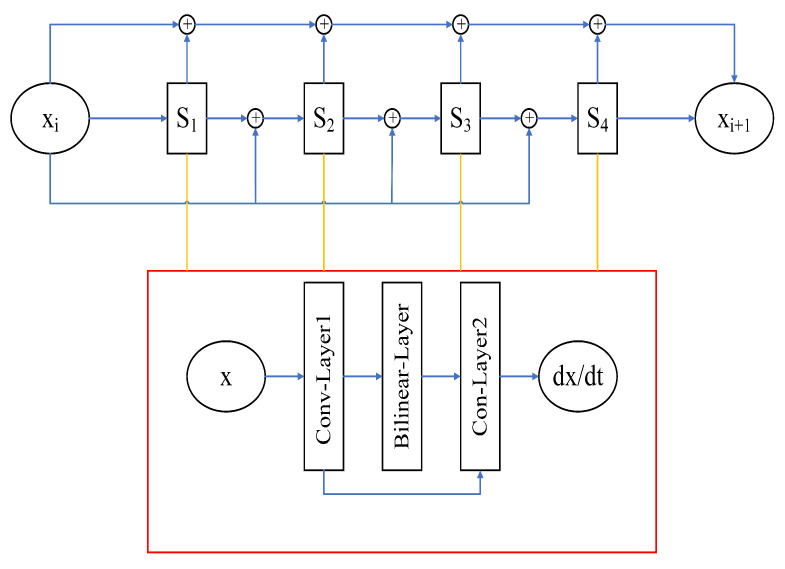
The architecture of the forecast model BNN.

**Figure 3 entropy-24-00264-f003:**
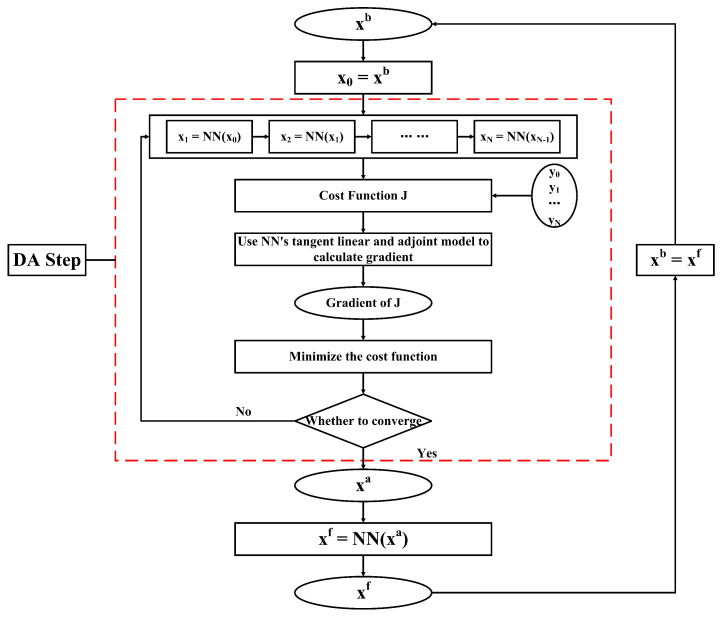
Schematic diagram of the ML-4DVAR.

**Figure 4 entropy-24-00264-f004:**
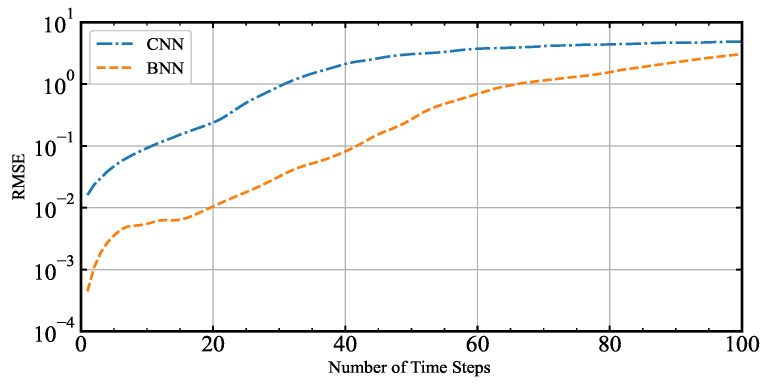
Comparison of the BNN and the CNN simulation effects.

**Figure 5 entropy-24-00264-f005:**
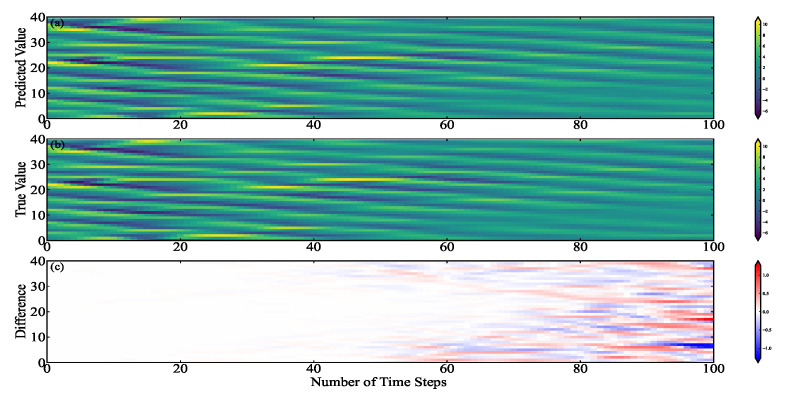
The temporal and spatial distribution of the output values of the BNN and the Lorenz-96 model under the same initial conditions, (**a**) the output of the BNN, (**b**) the output of the Lorenz-96 model, and (**c**) the difference between the two models.

**Figure 6 entropy-24-00264-f006:**
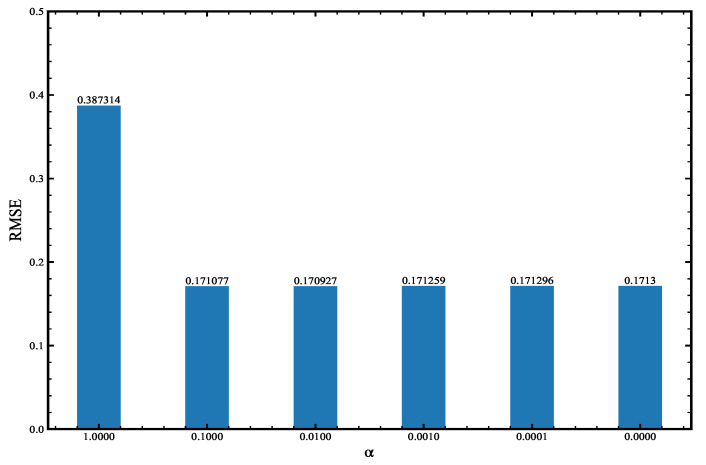
The value of RMSE under different α.

**Figure 7 entropy-24-00264-f007:**
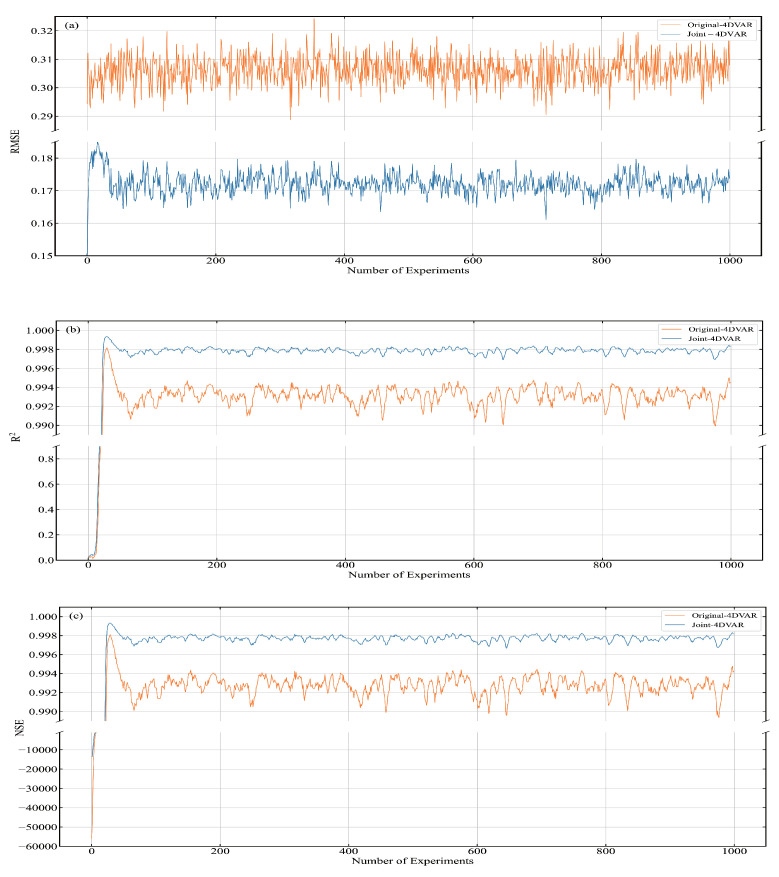
RMSE, $R^{2}$, and NSE of Joint-4DVAR and Original-4DVAR. (**a**) RMSE, (**b**) $R^{2}$, (**c**) NSE.

**Figure 8 entropy-24-00264-f008:**
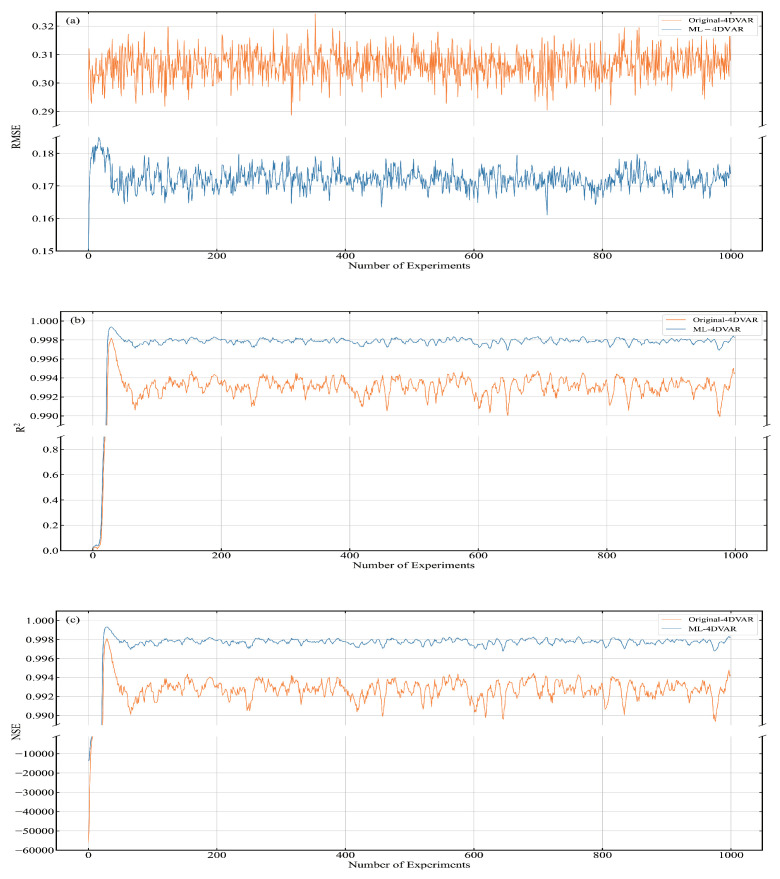
RMSE, $R^{2}$, and NSE of ML-4DVAR and Original-4DVAR at analysis time. (**a**) RMSE, (**b**) $R^{2}$, (**c**) NSE.

**Figure 9 entropy-24-00264-f009:**
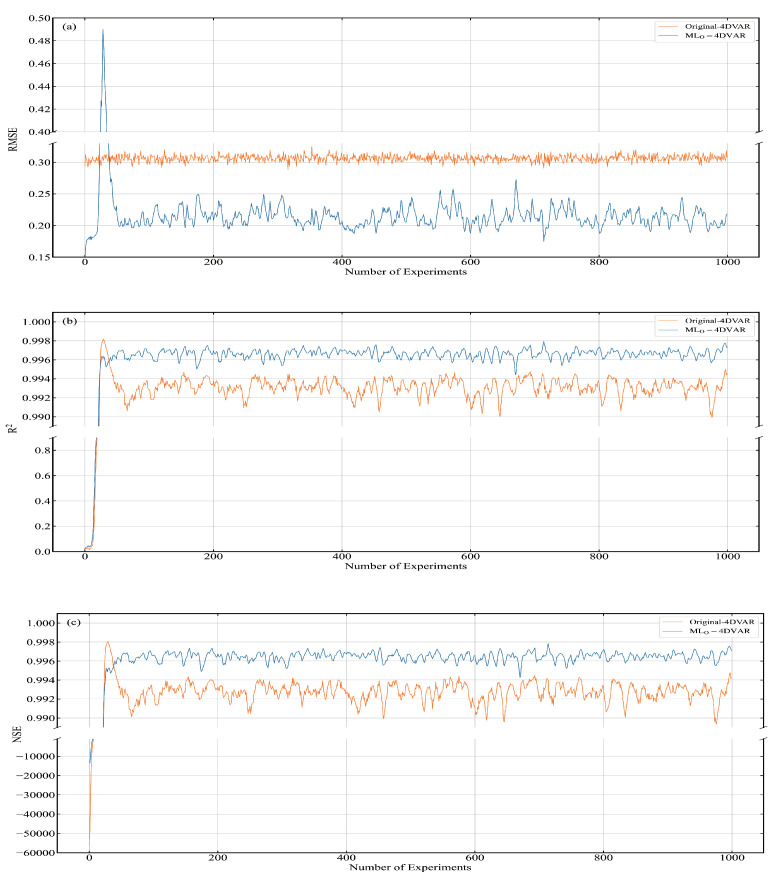
(**a**) RMSE, (**b**) $R^{2}$, and (**c**) NSE of ${\mathrm{ML}}_{O}$-4DVAR and Original-4DVAR.

**Table 1 entropy-24-00264-t001:** Comparison of the analysis of Joint-4DVAR and Original-4DVAR.

	RMSE	$R^{2}$	NSE
Joint-4DVAR	0.171965	0.997868	0.997706
Original-4DVAR	0.306383	0.993088	0.992716

**Table 2 entropy-24-00264-t002:** Comparison of the forecast of Joint-4DVAR and Original-4DVAR.

	RMSE	$R^{2}$	NSE
Joint-4DVAR	0.181307	0.997697	0.997452
Original-4DVAR	0.300698	0.993383	0.992985

**Table 3 entropy-24-00264-t003:** Comparison of the running time of Joint-4DVAR and Original-4DVAR (unit: s).

	Time
Joint-4DVAR	248.116567
Original-4DVAR	727.291506

**Table 4 entropy-24-00264-t004:** The assimilation performance and computational efficiency of ML-4DVAR and Original-4DVAR.

		RMSE	$R^{2}$	NSE	Time (Unit: s)
$x^{a}$	ML-4DVAR	0.169947	0.997871	0.997760	158.181050
	Original-4DVAR	0.306383	0.993088	0.992716	727.291506
$x^{f}$	ML-4DVAR	0.175781	0.997716	0.997605	158.181050
	Original-4DVAR	0.300698	0.993383	0.992985	727.291506

**Table 5 entropy-24-00264-t005:** The assimilation performance and computational efficiency of ML-4DVAR and Joint-4DVAR.

		RMSE	$R^{2}$	NSE	Time (Unit: s)
$x^{a}$	ML-4DVAR	0.169947	0.997871	0.997760	158.181050
	Joint-4DVAR	0.171965	0.997868	0.997706	248.116567
$x^{f}$	ML-4DVAR	0.175781	0.997716	0.997605	158.181050
	Joint-4DVAR	0.181307	0.997697	0.997452	248.116567

**Table 6 entropy-24-00264-t006:** The assimilation performance and computational efficiency of ${\mathrm{ML}}_{O}$-4DVAR and Original-4DVAR.

		RMSE	$R^{2}$	NSE	Time (Unit: s)
$x^{a}$	${\mathrm{ML}}_{O}$-4DVAR	0.213248	0.996640	0.996481	173.746833
	Original-4DVAR	0.306383	0.993088	0.992716	727.291506
$x^{f}$	${\mathrm{ML}}_{O}$-4DVAR	0.285967	0.993881	0.993653	173.746833
	Original-4DVAR	0.300698	0.993383	0.992985	727.291506

**Table 7 entropy-24-00264-t007:** The assimilation performance and computational efficiency of ${\mathrm{ML}}_{O}$-4DVAR and ML-4DVAR.

		RMSE	$R^{2}$	NSE	Time (Unit: s)
$x^{a}$	${\mathrm{ML}}_{O}$-4DVAR	0.213248	0.996640	0.996481	173.746833
	ML-4DVAR	0.169947	0.997871	0.997760	158.181050
$x^{f}$	${\mathrm{ML}}_{O}$-4DVAR	0.285967	0.993881	0.993653	173.746833
	ML-4DVAR	0.175781	0.997716	0.997605	158.181050

## Data Availability

The data and code of this study can be obtained by contacting the author, email address: dongrz20@nudt.edu.cn.

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
