# Peer review of "A Framework for Four-Dimensional Variational Data Assimilation Based on Machine Learning"

_entropy, 2022, doi:10.3390/e24020264_

Round 1

Reviewer 1 Report

References are incomplete, many basic works on using NNs for emulating numerical model components are missed.

Line 145; Authors stated that: “In theory, the neural network can fit any function [25],”

This statement is not proven yet.  It was proven that a shallow NN can fit any function (latest work that show this is: V. N. Vapnik, 2019. Complete Statistical Theory of Learning. Automation and Remote Control, Vol. 80, No. 11, pp. 1949–1975).  There is no such a general proof for DNNs.  Also, the reference [25] is not complete, or there is no such a paper at all.

Line 153: “to simulate the operator f to get f^”, at this point  f^ has not yet been introduced.

Lines 294-296; “The observation required by these three systems are the same, and they are all acquired by adding disturbances to the real values.” What real values are used, what disturbances are introduced? Are perturbed Lorenz-96 data called “observations”?

Line 464-465; “The method is more applicable. This method is used in more complex atmospheric models to obtain accurate forecast results.” It is not clear what the authors want to say. Is this method really used in more complex atmospheric models? Where and when?

Reviewer 2 Report

In the paper, Machine Learning is used for solving 4D variational data assimilation problems. Here, the original physically-based numerical model is replaced by a Bilinear Neural Network (BNN). Once this BNN is available, it becomes possible to use all the tools available for BNN's: The tangent linear model, the adjoint model, and the optimization algorithm. Tools that are necessary for solving variational data assimilation problems.

The major assumption for applying the methodology is that it is possible to derive a BNN that accurately approximates the simulations of the physically-based numerical model. This seems to me very hard for large scale, complex numerical models. And definitely not - as the authors are suggesting - by a BNN that is also computationally much faster than the physically-based numerical model. But there are probably also applications where this idea might be useful, like the application to the Lorentz-96 model described in the paper.

The paper is not well written and very hard to follow:

1) The English is not good. Many spelling mistakes, missing words (especially "the" or "a"), wrong sentences, sentences that are not complete, typo's. The paper makes a very messy impression.

2) Notation is not very precise. In equation (1) the model operator used is Mi from time 0 until time i, in equation (4) the model operator is M from time k until k+1, in equation (6) we suddenly have M-hat. And in equation (3) we have the model operator F for the dynamics of xk. All different notations for slightly different operators. The same holds for the time variable: time is i in equation (1) when it is about measurements, time is k when it is about time steps of the model in equation (4), and time can also be tk when it is about the complete time window of the assimilation. So I conclude the N in equation (1) must be the same quantity as tk+1 in the text after equation (6). And why is n introduced instead of i? 

3) In the section 3.2 there is a comparison with the CNN used by Seiya [34]. But this section is hard to understand without having read this paper.

4) The Original-4DVAR, Joint-4DVAR and ML-4DVAR are not clearly introduced and explained.

Concluding: A paper with an interesting idea, but not publishable in its present form. It should be rewritten completely, probably with the help of a native English speaker.

Round 2

Reviewer 1 Report

No new comments.

Reviewer 2 Report

The presentation has been improved a lot. There are still some minor errors with respect to the English language, but this does not prevent a good understanding of the paper. I advise accepting the paper.